# Polymers under Load and Heating Deformability: Modelling and Predicting

**DOI:** 10.3390/polym13030428

**Published:** 2021-01-29

**Authors:** Alexander Korolev, Maxim Mishnev, Dmitry Zherebtsov, Nikolai Ivanovich Vatin, Maria Karelina

**Affiliations:** 1Institute of Architecture and Construction, Department of Building Structures and Constructions, South Ural State University, 454080 Chelyabinsk, Russia; 2Nanotechnology Research & Education Centre, South Ural State University, 454080 Chelyabinsk, Russia; 3Institute of Civil Engineering, Peter the Great Saint Petersburg Polytechnic University, 195251 Saint Petersburg, Russia; 4Department of Machinery Parts and Theory of Mechanisms, Moscow Automobile and Road Construction University, 125319 Moscow, Russia; karelinamu@mail.ru

**Keywords:** polymer, deformability, modulus of elasticity, modulus of deformation, mineral additives, layered model, thermal load

## Abstract

The polymer deformability under load and heating is the determining factor in calculating reinforced polymer structures used under heating. Deformability–load/temperature relations make it possible to calculate temperature stresses and deformations in bearing cross-sections of polymer structures such as chimneys, smokestacks, etc. The present study suggests a method of calculating deformability of polymers subjected to the temperature loads. The method is based on the structure model of pack or layer bonded polymer domains where the elasticity of rigid bonds decreases with heating according to entropy principles. The method has been successfully tested on various polymers and compounds with due account for the effect of mineral additives on the deformation modulus increase.

## 1. Introduction

Deformability of polymer materials subjected to mechanical and temperature loads are determined by its structure specificities [1,2,3,4]. According to studies of the structure of glassed polymers [5,6,7,8,9,10], including those after thermal relaxation, the structure specificities are as follows:

The structure of glassed polymers is divided into molecular and supramolecular depending on the measurement level. Characteristics of molecules and molecule aggregates, their particle size and distribution in structure volume determine the deformative properties.

After glassing, the structure of polymers is a domain structure; the supramolecular structure consists of the bonded regular basic aggregates entropically oriented to one another. Regular macro levels in such a structure cannot be detected according to crystallization completeness. Depending on each author’s terminology for basic aggregates, the domain structure could also be called the block structure or pack structure. Its behaviour under mechanical load could be represented as resistance of layers or domains.

A domain-like structure is formed owing to separation of polymer molecules into fractions during glassing reaction. The most volatile and unreactive fractions are displaced on the peripheral surfaces of hardening domains. They form the border between phases of the interphase transition zone (ITZ). The ITZ includes mostly weakened areas and forms endless ways to replace volatile fractions on the outer layers of the polymer structure due to heating, dissolving, and other factors.

With temperature increase, polymer strength decreases, and deformability grows. It is related to reversible and irreversible processes in a structure under heating. Inter-domain bonds have less energy than inner-domain ones, so they are probably adsorptive or van-der-Waals ones [11,12,13,14]. Therefore, these bonds have an ability for irreversible deformations after mechanical or thermal energy impacts. In common, these processes have an entropy character and bring to an increase of relaxation and regularity of polymer structures resulting in thermal energy emission [15,16,17].

Finally, with heating, inter-domain bonds of polymer structure are losing elasticity, and more areas between domains are losing rigidity. It makes polymer deformability viscoelastic. Viscoelasticity is a base for modern modelling of polymer mechanical and deformative properties [18,19,20,21,22,23,24]. Many polymer deformability models are developed [25,26,27,28,29,30,31], but most of them have low technological applicability and do not predict polymer structure properties under load together with heating.

The presented research aims at generating improved polymer deformability calculation models that can consider temperature factor by using surface inter-domain elements in polymer structure forming and the thermodynamical entropic mechanism.

The object of research was a glassed polymer with epoxy, phenolic and epoxy-phenolic glassed binders by the example.

The subject of this research was the glassed polymer deformability under heating in relation to temperature change in the polymer structure.

The research includes several objectives:-Thermo-gravimetrical research of glassed polymers aimed at determining entropy change;-Experimental study of the polymer “deformation modulus-temperature” dependence;-Formulation and testing of the calculation model for layer-bonded polymer deformability depending on temperature factor.

## 2. Materials and Methods

The experimental research was conducted at the South Ural State University, Chelyabinsk, the Russian Federation.

### 2.1. Materials

The characteristics of thermosetting polymer materials are presented in Table 1. For these materials the thermal stability (as weight loss) and deformability were measured at elevated temperatures. The table shows the percentage by mass of components in the composition of the binder.

The components described below were used to make the binders.

Epoxy resin KER 828 with the following main characteristics: EGC 5308 mmol/kg, EEW 188.5 g/eq, viscosity at 25 °C 12.7 Pa·s, HCl 116 mg/kg, total chlorine 1011 mg/kg. Manufacturer KUMHO P&B Chemicals, Gwangju, South Korea.

Hardener for epoxy resin methyl tetrahydrophthalic anhydride with the following main characteristics: viscosity at 25 °C 63 Pa·s, anhydride content 42.4%, volatile fraction content 0.55%, free acid 0.1%. Manufacturer ASAMBLY Chemicals company Ltd., Nanjing, China.

Alkofen (epoxy resin curing accelerator) with the following main characteristics: viscosity at 25 °C 150 Pa·s, molecular formula C_15_H_27_N_3_O, molecular weight 265, amine value 600 mg KOH/g. Manufacturer Epital JSC, Moscow, Russian Federation.

Resol phenolic resin SFRZ-309 with the following main characteristics: viscosity at 25 °C 700 mPa·s, not more than 20% (m/m) water, not more than 20% (m/m) free phenol. Manufacturer FCP “Sverdlov Plant”, Dzerzhinsk, Russian Federation.

Dispersed filler micro-grained marble (microcalcite) with the following main characteristics: not less than 98% CaCO_3_, not more than 0.3% substances insoluble in hydrochloric acid, the number of particles less than 2 μm is 15%, the median average particle diameter is 50 μm, the median maximum particle diameter is 98 μm. Manufacturer Koelgamramor Ltd., Koelga, Russian Federation.

The components were mixed in the above proportions at room temperature of about 25 °C. Mixing to a homogeneous consistency was conducted mechanically with an electric drill with a mixing attachment.

Micro-grained marble was preliminarily dried at a temperature of 110 °C and sieved through a sieve with a mesh of 0.06 mm to separate the aggregates.

### 2.2. Methods

The thermogravimetric (TG) investigation was done using a simultaneous thermal analyser Netzsch STA 449C “Jupiter” (Selb, Germany) in the argon atmosphere in corundum crucibles with the sample mass 10–15 mg and size about 1–2 mm in cross-section. During TG test the sample mass, heat absorption and emission, temperature under heating were detected with producing the TG curves, as well as the differential TG (DTG) and differential scanning calorimetry (DSC) curves. 

The sample morphology was explored using a Schottky emission scanning electron microscope Jeol JSM-7001F (Tokyo, Japan).

Three-point bending tests of polymer samples were carried out on a Tinius Olsen h100ku testing machine (Horsham, USA) in a specially made small-sized chamber, which provides heating and maintaining the temperature up to 300 °C.

According to the producer’s data for a Tinius Olsen h100ku machine, the load accuracy was ±0.5% in the range 0.2–100% of the installed force sensor (100 kN). The resolution of measuring the crosshead movement was 0.1 mm, with an error of up to 0.01 mm. The sample centre point displacement under the load was monitored by a mechanical dial gauge mounted on the bottom of the small-sized test chamber. This monitoring was aimed at excluding the machine compliance influence. The difference between the displacement readings along the traverse and the dial gauge did not exceed 2%.

Three-point bending tests determined the cured sample deformation modulus at temperatures from 22 to 150 °C. The tests were conducted following Russian State Standard GOST 25.604-82 [32].

Binders were poured into silicone moulds to make the experimental samples. The sample sizes: length 80 mm, width 10 mm (±1.0 mm), thickness 4 to 5 mm. The thickness of the samples was variable as they were machined (ground) to remove surface defects. The actual dimensions of the tested samples were measured with a calliper with an accuracy of 0.01 mm.

Samples based on epoxy resin were cured in an oven at a temperature of 150 °C for 30 min. Samples based on phenolic and epoxy-phenolic resins were cured at a temperature of 90 °C for 6 to 10 h.

The experimental values of elasticity modulus at bending of the samples were determined at a 2 mm/min loading rate. The determination of the elasticity modulus was conducted under loading with two load steps.

During tests conducted at temperatures up to 80 °C, the samples at the first (preliminary) stage were loaded with a concentrated force of 5 N (0.5 kgf), and then at the second stage of loading with a concentrated force of 10 N (1 kgf). The elasticity modulus was determined at the second stage of loading.

During the tests conducted at a temperature of 80 to 100 °C, the samples at the first (preliminary) stage were loaded with a concentrated force of 3 N (0.3 kgf), and then at the second stage of loading with a concentrated force of 7 N (0.7 kgf). The elasticity modulus was determined at the second stage of loading.

During tests conducted at temperatures above 100 °C, the samples at the first (preliminary) stage were loaded with a concentrated force of 1.5 N (0.15 kgf), and then at the second stage of loading with a concentrated force of 2.5 N (0.25 kgf). The elasticity modulus was determined at the second stage of loading.

The experimental values of the elasticity modulus for each type of studied binders were obtained from the results of testing three samples. A general view of a silicone mould for making samples and cured samples of binders is shown in Figure 1.

The temperature during the tests was maintained by a thermostat and controlled by two thermocouples. One thermocouple measured the temperature on the surface of the bent specimen. The second thermocouple measured the temperature inside the control specimen, located next to the test specimen. The installation diagram is shown in Figure 2.

## 3. Results and Discussion

### 3.1. Layer-Bonded Polymer Deformability Calculation Model Formed on the Basis of the Research in Polymer Elasticity and Entropy under Heating

Experimentally defined dependences of epoxy, epoxy-phenolic, and phenolic polymers deformation moduli on temperature are shown in Figure 3 and Figure 4; they demonstrate the intense monotonically decreasing modulus with increasing temperature. The modulus temperature breaking point was minimal for epoxy resin (95–100 °C), and it increased with phenolic content. 

The mass–emission and heat–emission of the glassed polymers with temperature increase (Figure 5) were studied to evaluate structural changes in polymers.

Characteristic feature in curves is the simultaneousness of the heat emission and heat absorption processes (DSC curves) with change in temperature. In the case when the dominating heat absorber is the polymer heat capacity, heat absorption is proportional to increasing temperature. However, the dependence is actually disproportional. It can be related with simultaneous overlaying of such physico-chemical processes as sublimation, adsorption, and crystallization on the physical heat capacity. Heat emission significantly increases in the temperature range between 100 to 150 °C. With the increase in mass loss (TG curve), heat emission increases, too. Increase of the heat emission process with temperature points out that the inner structure is self-regulating and energy releasing, and entropy-changing processes are taking place because of polymer structure changing, accompanied by irreversible heat emission.

The heat emission process can be attributed to a crystallization process, but this is not clearly supported by the following facts:

The mass loss process is not going with macro destruction, and porous capillary structure in polymer structure is not determined;

The crystallization process is not related to mass loss, and the actual simultaneous mass loss is not explained.

The surface effects determine the formation of the polymer structure; accordingly, the structure is conglomerate and consists of crystallized domain layers bonded by the adsorption interactions formed by the adhesive forces on the interphase/interdomain boundary. Under heating, adsorptive bonds of internal liquid (non-glassed monomers, solvents, volatile fractions) molecular layers weaken, and volatile, or decomposition, products sublimate. As a result, the distance between layers shortens, and molecular forces increase with heat emission. In support of this assumption, the DSC curves of polymers containing a solvent (acetone) show that heat emission increases significantly. Thus, the solvent compensates interphase energy and, after sublimation, releases more heat energy detecting greater initial entropy.

Therefore, the thermal balance of the thermo-isolated heated glassed polymer can be presented as a sum of inner energy and entropy of the polymer structure
(1)Q=Cm+TΔS

C is the relative heat capacity of glassed polymer;

m is the mass of a polymer;

T is the absolute temperature;

ΔS is the entropy of the glassed polymer.

The entropy change is related to modifying the structure regularity that can lead to differing intermolecular and interlayer distance at that measurement level r_0_. Thus, entropy change can be determined as
(2)$$\Delta {S=C}_{r}\ln\frac{r_{t}}{r_{0}}$$
$$r_{t}\sim T$$

If included, interphase liquid:(3)$$\frac{r_{t}}{r_{0}}=\frac{V_{t}/S_{r}}{V_{0}/S_{r}}=\frac{m_{t}/\rho }{m_{0}/\rho }=\frac{m_{t}}{m_{0}}$$

ρ, $V_{0}$, $V_{t}$, $m_{0}$, $m_{t}$ are density, volume, and mass of interphase liquid before and after heating;

$S_{r}$ is a relative interphase area.

If distance shortens, the conditions for heat emission appear, and entropy take a negative sign.

In this case, much of the volatile fractions sublimate from the polymer structure, which results in thermo-relaxation, the regularity process increases after repeated heating; this is related to prolonged sublimation of the volatiles bonded and closed between crystallized layers. Therefore, the heating of severed layers results in shortening interlayer distances, because of thermal expansion and sublimation of volatiles. Thus, the entropy process is simultaneously related with thermal and mass changes, accompanied by heat emission; entropy has a negative sign and in relation to temperature can be expressed as
(4)$$\Delta S=-C_{T}\ln\frac{T_{s}}{T_{0}}-C_{m}\ln\frac{{\Delta m}_{\max}}{{\Delta m}_{\max}-{\Delta m}_{T}}$$

$C_{T}$ is the relative thermal entropy on the thermo-relaxation stage;

$C_{m}$ is the relative mass entropy on the thermo-relaxation stage.

Then,
(5)$$Q=\mathrm{Cm}\left(T-T_{0}\right)-C_{T}\ln\frac{T_{s}}{T_{0}}-C_{m}\ln\frac{{\Delta m}_{\max}}{{\Delta m}_{\max}-{\Delta m}_{T}}$$

${\Delta m}_{T}$ is the relative mass lost under actual temperature;

${\Delta m}_{\max}$ is the maximal mass loss under heating.

The actual and calculated data of heat emission correlate. The calculation is
(6)$$Q=-{11.1\;\mathrm{mV}}_{t}/g\;\left(t-25\;{}^{\circ}C\right)+C_{T}\ln\frac{{273K+t}_{s}}{298K}+C_{m}\ln\frac{{\Delta m}_{\max}}{{\Delta m}_{\max}-k_{a}{\;\Delta m}_{T}}$$

$k_{a}$ is the coefficient of sublimation allowing for the rate of sublimation from the outer surface.

Testing has proven the applicability of the polymer heat emission model. Entropy parameters during thermo-relaxation have been determined by this calculation method for different compounds (Figure 6).

The resulting data proves the assumption about adsorptive forces domination in the supramolecular structure interlayer bonds:

Solvent addition results in a total entropy increase, especially thermal, leads to separating polymer layers. After heating and sublimation, it results in releasing surface energy, and more intense heat emission.

Filler addition decreases thermal entropy and increase mass entropy. It is related to density increasing with introduced mineral particles.

Epoxy-phenolic resins are characterized by minimal entropy level. It is related to the significant interaction of different molecules and maximal structure regulation during the glassing process.

The layers of domains of the glassed epoxy polymer supramolecular structure with a micro-grained marble (or microcalcite) filler, which are similar to each other, can be detected on electron microphotography.

Detailed areas represent the level of regularity of inner domain packs near 1-micrometer size (Figure 7). Cleavage surfaces, which are ITZ surfaces, are detected between the blocks.

The proposed bonded model describes the glassed polymer structure as a pack of elastic layers severed by the interphase adsorptive-bonded organic liquid. At these conditions, this liquid has properties of a solid but with less elasticity than the basic elastic layers. This model, with decreasing elasticity of ITZ under load and heating, includes the following conditions:

The model is a conglomerate consisting of equal-elasticity layers severed by and interacted over adsorptive-bonded layers of lesser elasticity, forming ITZ. The layer consists of the oriented polymer domains which work as a layer in the direction perpendicular to thermo- or mechanical load.

ITZ elasticity modulus intensely decreases with heating, which makes the total elasticity of the glassed polymer structure decrease. After the critical temperature is achieved, ITZ loses elasticity entirely, and the structure becomes pseudoplastic, where elastic layers are able to shift on each other’s interphase surfaces.

ITZ elasticity decrease is related to decreasing bond energy, and structure entropy increases with heating.

Figure 8 presents the layer-bonded model scheme.

### 3.2. Formulation and Testing of Layer-Bonded Model in Polymer Elasticity Modulus in Relation with Temperature and Entropy Factor

According to the layer model, the fraction of elastic layers in a structure is n, and the fraction of ITZ layers with variable decreasing elasticity is (1 − n). With temperature T increase, the ITZ elasticity modulus decreases. If stresses in elastic layers with an elasticity modulus $E_{\mathrm{el}}$ and ITZ deformability modulus $E_{\mathrm{pl}}$ were equal, then deformations appearing under load $\varepsilon_{\mathrm{el}}$, $\varepsilon_{\mathrm{pl}}$ sum into total deformation $\varepsilon_{t}$. In that case, the polymer modulus of deformation is
(7)$$E_{t}=\frac{\sigma_{t}}{\varepsilon_{t}}=\frac{\sigma_{t}}{\varepsilon_{\mathrm{el}}+\varepsilon_{\mathrm{pl}}}=\frac{\sigma_{t}}{n\frac{\sigma_{t}}{E_{\mathrm{el}}}+\left(1-n\right)\frac{\sigma_{t}}{E_{\mathrm{pl}}}}=\frac{E_{\mathrm{el}}E_{\mathrm{pl}}}{{(1-n)E}_{\mathrm{el}}{+\mathrm{nE}}_{\mathrm{pl}}}$$

The shift coefficient is introduced as:(8)$$k_{\mathrm{pl}}=\frac{E_{\mathrm{pl}}}{E_{\mathrm{el}}}$$

If deformation modulus under a temperature 293 K is supposed to be equal to elasticity modulus $E_{\mathrm{el}}$ = $E_{0}$, then modulus of deformation depending on temperature is
(9)$$E_{T}=\frac{k_{\mathrm{pl}}E_{0}}{{1-n}_{T}{(1-k}_{\mathrm{pl}})}$$

Basing on previous temperature-changing polymer structure research results, let us propose that elasticity of adsorptive bond is related to temperature by the entropy equation, because entropy is a non-regularity measure, therefore,
(10)$$k_{\mathrm{pl}}=1-S\times \ln\frac{T_{s}}{T_{0}}$$
S [J/J] is the coefficient of the bond entropy equalling to the relation between entropy and potential energy of elastic bond at standard temperature.

Let us correlate the proposed Equations (3) and (10) with actual data of deformation modulus under heating. For epoxy resin composition the equation is
(11)$$k_{\mathrm{pl}\text{-}\mathrm{epox}}=1-3.1\ln\frac{{273+t}_{s}}{298}$$
(12)$$E_{T}=\frac{k_{\mathrm{pl}\text{-}\mathrm{epox}}E_{0}}{1-{0.77(1-k}_{\mathrm{pl}\text{-}\mathrm{epox}})}$$

For phenolic resin composition
(13)$$k_{\mathrm{pl}\text{-}\mathrm{fenol}}=1-1.45\ln\frac{{273+t}_{s}}{298}$$
(14)$$E_{T}=\frac{k_{\mathrm{pl}\text{-}\mathrm{fenol}}E_{0}}{{1-0*(1-k}_{\mathrm{pl}\text{-}\mathrm{fenol}})}=k_{\mathrm{pl}}E_{0}$$

For epoxy-phenolic resin composition
(15)$$k_{\mathrm{pl}\text{-}\mathrm{ep}\text{-}\mathrm{fen}}=1-2.1\ln\frac{{273+t}_{s}}{298}$$
(16)$$E_{T}=\frac{k_{\mathrm{pl}\text{-}\mathrm{ep}\text{-}\mathrm{fen}}E_{0}}{{1-0*(1-k}_{\mathrm{pl}\text{-}\mathrm{ep}\text{-}\mathrm{fen}})}=k_{\mathrm{pl}}E_{0}$$

This means that phenolic and epoxy-phenolic polymers do not have the layer structure. Its entropy is 2 and 1.5 times more than the epoxy composition entropy.

Table 2 presents the actual and calculated instant deformation modulus data. Accordingly, in the operating temperature range up to 120 °C for epoxy polymer and 200 °C for phenolic polymer, where elasticity is significant, the calculated and actual data agreement is high. Calculated and actual values difference is not more than 15%.

Taking into account filler addition, the layer model leads to
(17)$$E_{c}=\frac{E_{\mathrm{epox}}E_{\mathrm{mr}}}{{(1-n)E}_{\mathrm{epox}}{+\mathrm{nE}}_{\mathrm{mr}}}$$
where n is the relative volume fraction of a resin in the composition.

For example, deformation modulus of epoxy polymer with micro-grained marble composite in 1:1 ratio is
$$E_{\mathrm{epox}+\mathrm{mm}}=\frac{2953\times 70,000}{(1-0.5)\times 2953+0.5\times 70,000}=5666.9\;\mathrm{MPa}$$

## 4. Conclusions

As a result of analytical and experimental study, the applicability of the layer-bonded model with thermodynamical parameters has been proved in the calculation of deformation modulus depending on temperature under heating. The model operates the concepts of adsorptive bonds between the supramolecular layers in the conglomerate polymer structure, and these bonds are highly sensitive to temperature increase.

The model differs from others even those using entropy factor, in that it is based not on statistical principles, but on energy effects. Structure representation is available easily enough to provide accurate calculations. It includes the ITZ layer as a principal element, except elastic polymer layers.

Model factors are ITZ elasticity under heating, entropy coefficient of adsorptive bonds for a specific polymer, “cold” elasticity modulus of a glassed thermo-relaxed polymer, and the relation between the domain and ITZ layer sizes.

Application of this model can be useful in calculating stresses in heated polymer-reinforced structures and developing methods of heat resistance increase of polymer composites.

## Figures and Tables

**Figure 1 polymers-13-00428-f001:**
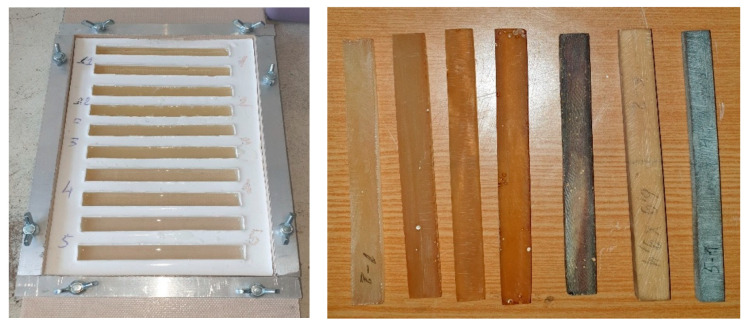
Samples of formed and glassed polymer compounds.

**Figure 2 polymers-13-00428-f002:**
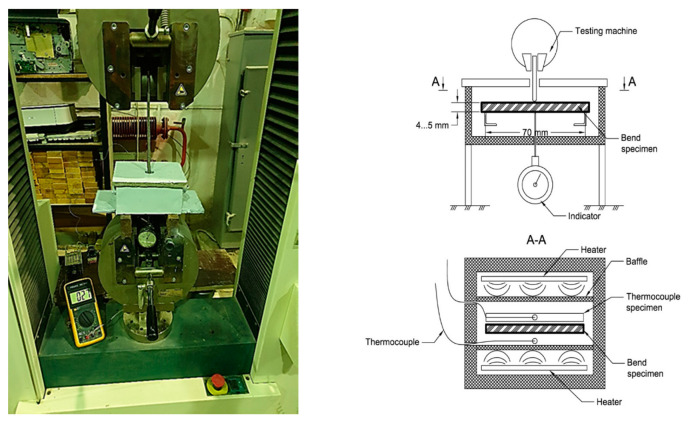
Heating Three-Point Bending Test Rig.

**Figure 3 polymers-13-00428-f003:**
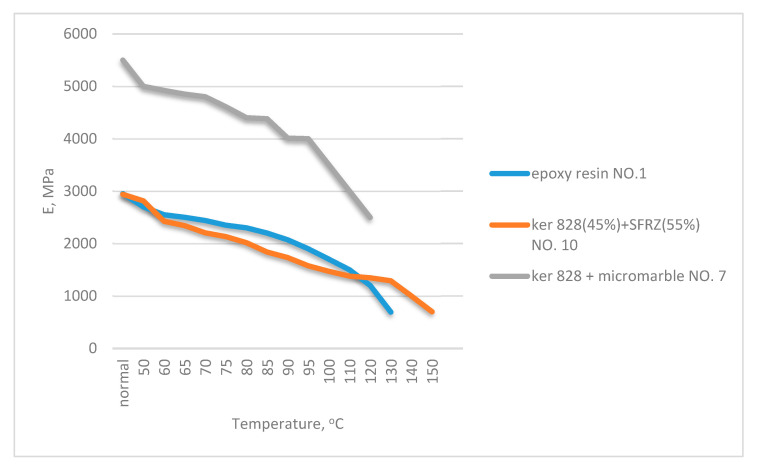
Dependence of the deformation moduli of epoxy and epoxy phenolic binders on temperature.

**Figure 4 polymers-13-00428-f004:**
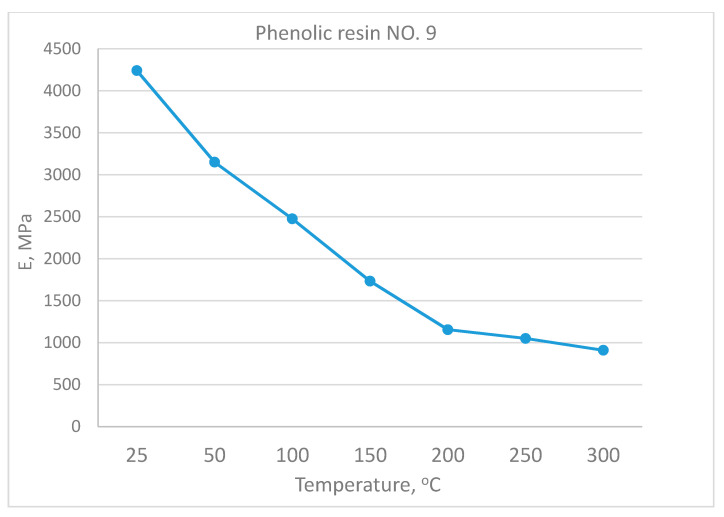
Dependence of the deformation moduli of phenolic resin on temperature.

**Figure 5 polymers-13-00428-f005:**
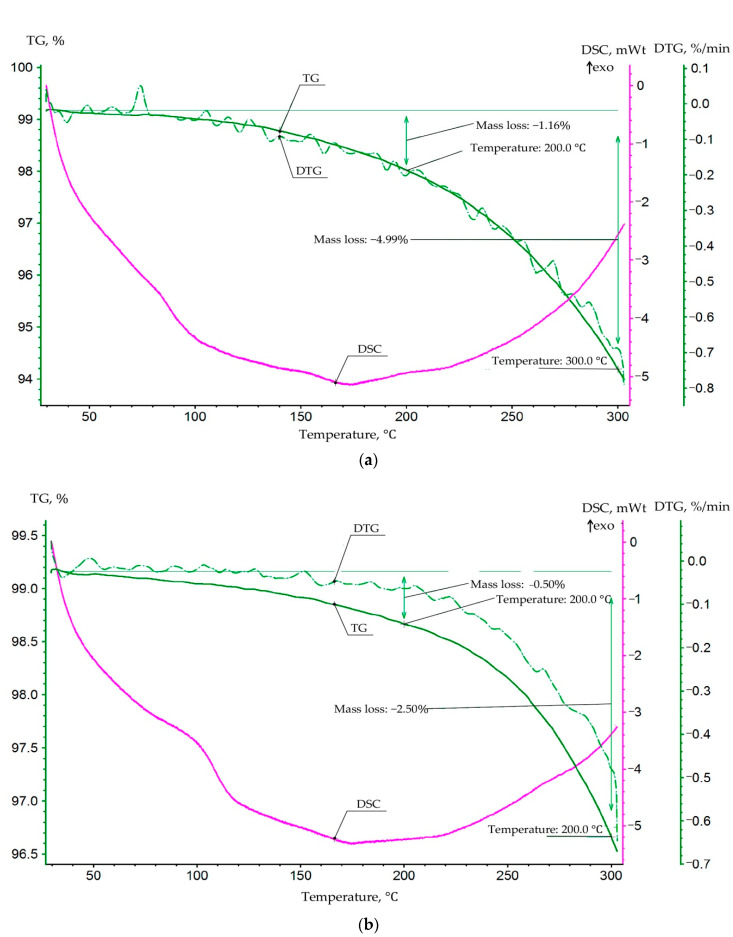
Thermogravimetric (TG), DTG, DSC curves of glassed polymers. (**a**) Epoxy resin; (**b**) epoxy resin with 5% acetone.

**Figure 6 polymers-13-00428-f006:**
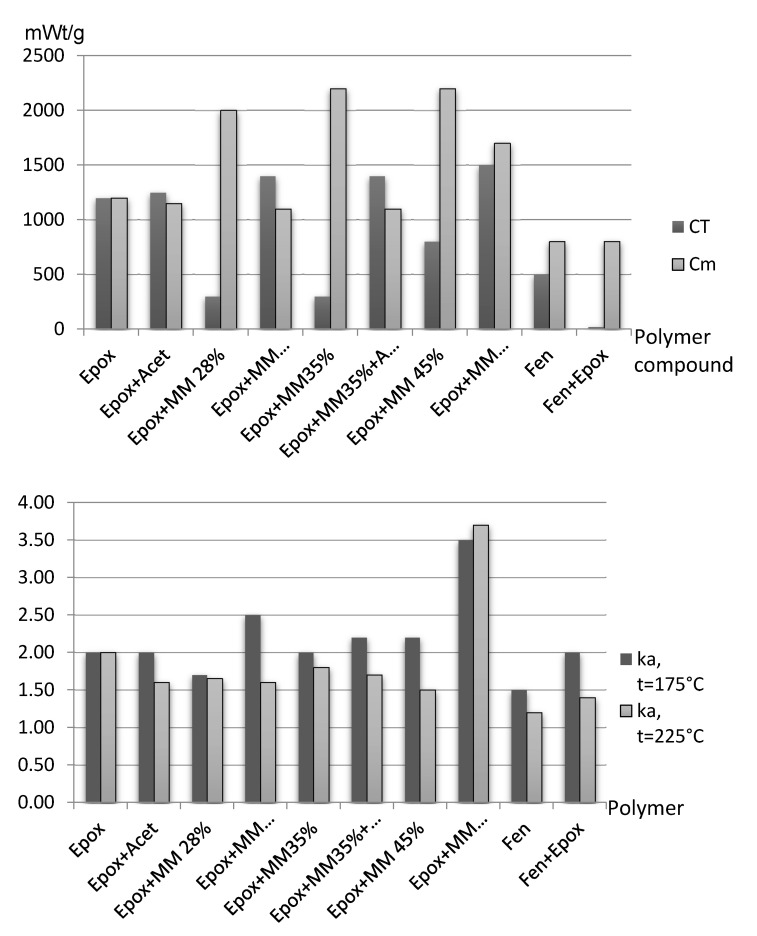
Calculated thermal/mass entropy characteristics and sublimation coefficient of glassed polymers and compounds under heating.

**Figure 7 polymers-13-00428-f007:**
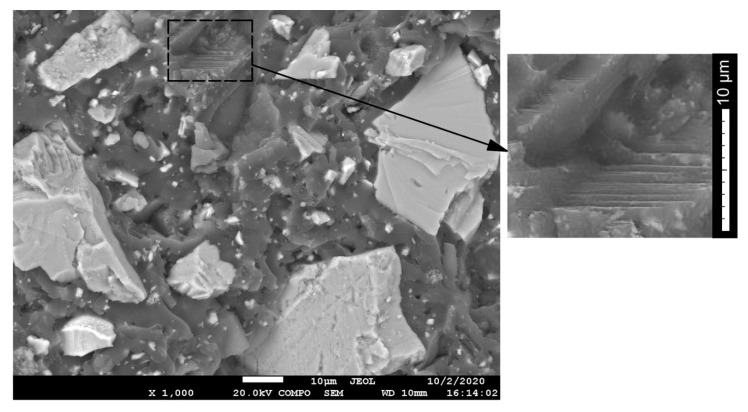
The layer-bonded structure of micro-filled epoxy polymer.

**Figure 8 polymers-13-00428-f008:**
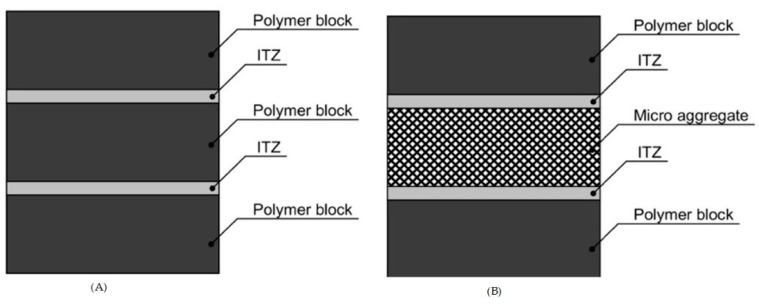
Layer-bonded model of polymer deformability. (**A**) Polymer domain model, (**B**) Polymer domain model with micro aggregate.

**Table 1 polymers-13-00428-t001:** Types of binders investigated.

No.	Compound	Thermo-Gravimetric Analysis	Three-Point Bending Test at Elevated Temperature
1	Epoxy resin (Ker 828 52.5% + MTHPA 44.5% + alkofen 3%)	+	+
2	Epoxy resin (No. 1) 81% + acetone 9%	+	-
3	Epoxy resin (No.1) 72% + micro-grained marble 28%	+	-
4	Epoxy resin (No. 1) 72% + micro-grained marble 24% + acetone 4%	+	-
5	Epoxy resin (No.1) 65% + micro-grained marble 35%	+	-
6	Epoxy resin (No. 1) 62% + micro-grained marble 33% + acetone 5%	+	-
7	Epoxy resin (No.1) 55% + micro-grained marble 45%	+	+
8	Epoxy resin (No. 1) 48% + micro-grained marble 48% + acetone 4%	+	-
9	Phenolic resin (SFZ-309) 100%	+	+
10	Epoxy-phenolic resin (ker 828 45% + SFZ-309 55%)	+	+

**Table 2 polymers-13-00428-t002:** Comparative actual and calculated data for modulus of deformation.

Compound	Temperature, °C	E_fact_, MPa	S, J/J	**K** _**pl**_	E_calc_, MPa	% Derivation
Epoxy No. 1	25	2953	3.1	1.000	-	-
	50	2700		0.750	2743.0	1.6
	75	2350		0.519	2434.4	3.6
	100	1700		0.304	1934.7	13.8
	130	693		0.064	679.2	−2.0
Epoxy + micro-grained marble No. 7	25	5500	3.1	1.000	5666.9	3.0
	50	5000		0.750	5108.9	2.2
	75	4610		0.519	4534.1	−1.6
	100	3500		0.304	3603.4	3.0
	120	2500		0.142	2303.5	−7.9
Phenolic No. 9	25	4242	1.45	1	-	-
	50	3150		0.883	3746.4	18.9
	100	2475		0.685	2909.7	17.5
	150	1732		0.509	2162.8	24.8
	200	1155		0.353	1499.2	29.8
	250	1050		0.212	902.2	−14.0
	300	909		0.084	359.5	−60.4
Epoxy-phenolic No. 10	25	2940	2.1	1	-	-
	50	2814		0.831	2442.6	−13.2
	75	2131.5		0.674	1982.4	−7.0
	100	1470		0.529	1554.0	5.7
	125	1347.2		0.419	1231.6	−8.6
	150	700		0.264	777.4	11.1
	175	-		0.144	422.9	-

## Data Availability

The data presented in this study are available on request from the corresponding author.

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
