# Peer review of "Polymers under Load and Heating Deformability: Modelling and Predicting"

_polymers, 2021, doi:10.3390/polym13030428_

Round 1
Reviewer 1 Report
This paper presented an investigation of polymer deformation under heating and external load conditions. The authors first briefly reviewed the state of the art. Then the materials and testing procedures were introduced. The authors reported some results to support their discussion. However, this paper is not well organized. The reviewer suggests a mandatory revision before publishing this paper.
- The introduction is brief and doesn't cover the important previous work related to the topic. Please discuss the key milestones of the published work in this field and clearly present the novelty of this paper in the revised paper.
- In the Materials and Methods section, please clarify if the mechanical tests followed any standards, such as ISO standard for 3 point bending. If so, what is the sample size used in this study. If not, what is the loading/unloading procedures used in this paper?
- Please clarify that if multiple tests were conducted to verify the repeatability of this study.
- There are multiple grammatical errors in the current draft. Please proofread this paper before submitting the revised paper.
Author Response
|
Dear Reviewer, The authors would like to express their sincere gratitude for the additional comments that have improved the work. Many comments and recommendations will also be taken into account in future studies. |
|
|
The authors reported some results to support their discussion. However, this paper is not well organized. The reviewer suggests a mandatory revision before publishing this paper.
|
The article is fully reorganized. All experimental data distributed in Results and Discussion, there were introduced subitems for dividing materials and methods, experimental results and modelling with testing |
|
1) The introduction is brief and doesn't cover the important previous work related to the topic. Please discuss the key milestones of the published work in this field and clearly present the novelty of this paper in the revised paper.
|
In introduction research aim, object, subject and tasks were cleared. It made possible to focus novelty, research and previous works diapason |
|
2) In the Materials and Methods section, please clarify if the mechanical tests followed any standards, such as ISO standard for 3 point bending. If so, what is the sample size used in this study. If not, what is the loading/unloading procedures used in this paper?
|
This information added at lines 111-112. |
|
3) Please clarify that if multiple tests were conducted to verify the repeatability of this study
|
This information added at lines 113-114. |
|
4) There are multiple grammatical errors in the current draft. Please proofread this paper before submitting the revised paper.
|
The authors have done an additional language correction. |
Reviewer 2 Report
Dear Authors:
The manuscript has an interesting approach, because the understanding of the thermomechanical properties of polymers are a challenge for predicting their behavior under extreme conditions. Besides, the theoretical consideration are consistent and the results are valuable. However, the wording of the manuscript is confusing; I strongly recommend to improve it. Additionally, consider the following comments:
1. Figure 4 and 5 should be included and cited in results section.
2. Figure 6, the DSG acronym should be defined. Additionally, analysis details should be included in the methodology section.
Author Response
|
Dear Reviewer, The authors would like to express their sincere gratitude for the additional comments that have improved the work. Many comments and recommendations will also be taken into account in future studies. |
|
|
However, the wording of the manuscript is confusing; I strongly recommend to improve it.
|
The article is fully reorganized. The authors have done an additional language correction. |
|
1) Figure 4 and 5 should be included and cited in results section.
|
It’s done exactly (from line 124) |
|
2) Figure 6, the DSG acronym should be defined. Additionally, analysis details should be included in the methodology section.
|
This information added at lines 95-96, 142-146. |
Round 2
Reviewer 1 Report
The authors have revised the paper according to the comments given by the reviewer. The quality of the paper has been improved and the reviewer suggests publishing this paper.
Author Response
Dear Reviewer,
The authors once again express their gratitude for the helpful comments that helped improve the article.
Reviewer 2 Report
Dear Author,
In my opinion your work should be accepted after some changes will be introduced and some theoretical concerns will be clarify.
- page 2, line 57, the word "presenter" should be replaced by presented.
- page 2, line 72, the word "model" should have capital letter. The sense of the sentence is not clear, maybe the sentence can be replaced by Materials, Methods and Models
- The sentence "under temperature, more than 100...150°C heat emission is dominant" may be replaced by "heat emission is a dominant process in the temperature range between 100 to 150°C. However, why the authors indicates the heat-emission as the process that is dominant, if the DSC plot shows a endothermal process in this temperature range? verify if the arrow and word ekzo is right (ekzo, maybe referred to exo), probably this induces to misunderstand the discussion results.
- The sentence in page 6, line 156, may be replaced by "the heat emission process can be attributed to a crystalization process, but this is not clearly supported by the following facts".
- Page 8, line 214, "received data" probably should be replaced by "resulting data"
Author Response
|
Dear Reviewer, The authors would like to express their sincere gratitude for the additional comments that have improved the work. |
|
|
1. page 2, line 57, the word "presenter" should be replaced by presented.
|
It is done (line 53). |
|
2. page 2, line 72, the word "model" should have capital letter. The sense of the sentence is not clear, maybe the sentence can be replaced by Materials, Methods and Models
|
The word "model" is cleared (line 65). |
|
3. The sentence "under temperature, more than 100...150°C heat emission is dominant" may be replaced by "heat emission is a dominant process in the temperature range between 100 to 150°C. However, why the authors indicates the heat-emission as the process that is dominant, if the DSC plot shows a endothermal process in this temperature range? verify if the arrow and word ekzo is right (ekzo, maybe referred to exo), probably this induces to misunderstand the discussion results.
|
It is corrected on the base of suggestion (line 166). |
|
4. The sentence in page 6, line 156, may be replaced by "the heat emission process can be attributed to a crystalization process, but this is not clearly supported by the following facts".
|
It is done (line 171). |
|
5. Page 8, line 214, "received data" probably should be replaced by "resulting data" |
It is done (line 233). |